# *Rosmarinus officinalis* and Skin: Antioxidant Activity and Possible Therapeutical Role in Cutaneous Diseases

**DOI:** 10.3390/antiox12030680

**Published:** 2023-03-09

**Authors:** Federica Li Pomi, Vincenzo Papa, Francesco Borgia, Mario Vaccaro, Alessandro Allegra, Nicola Cicero, Sebastiano Gangemi

**Affiliations:** 1Department of Clinical and Experimental Medicine, Section of Dermatology, University of Messina, 98125 Messina, Italy; 2Department of Clinical and Experimental Medicine, School and Operative Unit of Allergy and Clinical Immunology, University of Messina, 98125 Messina, Italy; 3Division of Haematology, Department of Human Pathology in Adulthood and Childhood “Gaetano Barresi”, University of Messina, 98125 Messina, Italy; 4Departement of Biomedical and Dental Sciences and Morphofunctional Imaging, University of Messina, 98168 Messina, Italy

**Keywords:** *Rosmarinus officinalis*, rosemary, skin, cutaneous disease, oxidative stress, ROS, carnosol, skin cancer, anti-aging, lymphoma

## Abstract

The rosemary plant, Rosmarinus officinalis L., one of the main members of the Lamiaceae family, is currently one of the most promising herbal medicines due to its pharmaceutical properties. This research aimed to evaluate the antioxidant role of Rosmarinus officinalis and its bioactive compounds on the skin, with a focus on the newly emerging molecular mechanisms involved, providing extensive scientific evidence of its anti-inflammatory, antimicrobial, wound-healing and anticancer activity in dermatological practice. The search was conducted on articles concerning in vitro and in vivo studies in both animals and humans. The results obtained confirm the antioxidant role of R. officinalis. This assumption derives the possibility of using R. officinalis or its bioactive elements for the treatment of inflammatory and infectious skin pathologies. However, although the use of rosemary in the treatment of skin diseases represents a fascinating line of research, future perspectives still require large and controlled clinical trials in order to definitively elucidate the real impact of this plant and its components in clinical practice.

## 1. Introduction

Phytotherapy, to take its most classic meaning, is the branch of medicine that uses plants in their entirety or their components for medical purposes to treat or prevent a plethora of diseases. A cornerstone of various medical traditions over the centuries, above all Ayurvedic medicine, traditional Chinese medicine, allopathic and naturopathic medicine [1], it is, more than ever, a current and much-investigated discipline, even in extraordinary scenarios such as the most recent pandemic period [2,3,4,5]. Considering its growth as a field worthy of scientific interest [6,7,8,9], the need to introduce guidelines for a better understanding of clinical indications, efficacy and safety profiles is arising [1,10,11]. Among the many plants and their active constituents studied in this unconventional setting [12,13,14,15,16], the rosemary plant, *Rosmarinus officinalis L*., one of the main members of the Lamiaceae family, cultivated in the Mediterranean basin for culinary use [17], is listed as one of the most promising. This is due to its increasingly established clinical utility [18] supported by the development of breakthrough methods for the targeted extraction of its bioactive metabolites [19]. Rosemary extracts are commercially available in Europe and the USA for use as natural antioxidants in the food industry. They have received Generally Recognized as Safe (GRAS) status from the US Food and Drug Administration. Among these extracts, the most notable, being of scientific interest, are rosmarinic acid, a derivative of caffeic acid [20,21]; phenolic diterpenes, of which carnosic acid and carnosol are the most medically relevant [22,23,24]; flavonoids such as diosmin [25]; and rosemary essential oil (EO), consisting of more than a hundred chemical compounds, the main molecules of which include 1,8-cineole, α-pinene, α-terpineol, verbenone, limonene, bornyl acetate, terpinolene and camphor [26,27]. Rosmarinic acid, beyond its anti-infectious, antioxidant, anti-inflammatory and immunomodulatory properties, has been extensively investigated in recent years for its anti-cancer activity on various apparently functionally disconnected molecular targets leading to various types of cancer. This has justified the increasing efforts of nanomedicine in the development of therapeutic delivery systems to improve its bioavailability [28]. The EO of *Rosmarinus officinalis* owes its clinical notoriety to its anti-inflammatory potential [26], as well as its antioxidant action, elicited mainly at the hepatic level [29]. Carnosic acid and carnosol, in addition to their antioxidant and anti-inflammatory potential, are clinically relevant for their lipid and glucose metabolism regulatory activity, which would justify their use in the treatment of diabetes mellitus and its complications [30,31]. An interesting and reasonably recent line of research has narrowed the field of interest to the phenolic compounds of *Rosmarinus officinalis*, investigating their therapeutic potential in various neurological disorders, including neurodegenerative diseases [32,33], prion diseases [34], cerebral ischemia [35], neuropathic pain [36] and encephalomyelitis [37]. Along the lines of what has already been widely discussed, the purpose of our review is to provide a detailed overview of the potential beneficial effects of *Rosmarinus officinalis L.* and, more specifically, of the wide range of its bioactive constituents on various dermatological diseases, with a special focus on its promising antioxidant role and the ambitious attempts to elucidate the pathogenetic molecular pathways involved.

## 2. Results and Discussion

### 2.1. Rosmarinus Officinalis and Antioxidant Action

Oxidative stress is the pathogenic primum movens of most cutaneous disorders, as the skin is the organ most widely and severely exposed to oxidative stress, despite the extensive endogenous and exogenous antioxidant system at its disposal [38]. For descriptive purposes, the causative agents of skin oxidative stress can be divided into exogenous and endogenous, including intracellular metabolic processes. The main exogenous pro-oxidant agents include ultraviolet (UV) light, environmental pollution and chronic psychological stress. The synergistic action of these factors accelerates the processes of pigmentation and skin aging [39]. The latter recognizes oxidative stress secondary to UV irradiation as the primary causal agent; hence, the need to coin the term photoaging. A greater contribution is made by UVA, since UVB, while participating in the damage, has limited penetration capacity into the epidermis and its cells, including keratinocytes and melanocytes above all [40]. The UV-induced oxidative damage of assorted intracellular structures can be direct or indirect. UVB, for the most part absorbed in the stratum corneum, is essentially accountable for the direct damage that sees DNA as its biologically most crucial molecular target [41]. Over the last two decades, the antioxidant potential of *Rosmarinus officinalis* and its bioactive constituents has been extensively investigated in both in vitro and in vivo studies, especially for its promising therapeutic effects on UV-induced photoaging, atopic dermatitis (AD) and pollution-induced skin aging. Relative to the aforementioned time frame, one of the first in vitro studies on this topic highlighted the interesting proportionality between protein glycation-inhibiting activity and antioxidant activity [42]. This functional synergy is said to be largely attributable to the polyphenolic compounds of various plant extracts, including *Rosmarinus officinalis*, paving the way for their prospective therapeutic use in diabetic complications and aging. Further confirmation of the antioxidant potential of this plant came from Ezzat et al., who emphasized the anti-wrinkle action of the defatted rosemary extract (DER), an effect attributable largely to rosmarinic acid, the main phenolic compound, but also to the diterpenes carnosic acid, carnosol and rosmanol [43]. Furthermore, the encapsulation of this extract in transferomes improves its skin penetrability. The most recent and relevant confirmation of the antioxidant action of rosemary’s phenolic compounds comes from an in vitro study evaluating the radical-scavenging and anti-aging activity of aqueous and ethanoic extracts of five phenolic-rich selected herbs, including *Rosmarinus officinalis*, which showed both the highest antioxidant activity and the most pronounced anti-elastase, anti-tyrosinase and anti-collagenase activity [44].

Very recent is the interest in the curative potential of rosemary in skin oxidative damage from pollution. In a randomized, double-blind, placebo-controlled study, Nobile et al. evaluated the antioxidant efficacy of the oral supplementation of four phenol-rich plants, including R. officinalis, in one hundred Caucasian and Asian women living in the polluted urban area of Milan [45]. It was found that long-term supplementation improved all clinical–biochemical parameters monitored, including increased skin elasticity, reinforced skin barrier function and a reduction in wrinkle depth and black spots in the enrolled patients [45]. Hoskin et al. demonstrated, for the first time, the effective action of the topical application of a gel based on hydroalcoholic rosemary extract complexed with algae proteins on pollution-induced oxidative skin damage [46]. The emerging molecular mechanisms through which rosemary phenolic diterpenes acts are the inhibition of the increasing levels of active metalloproteinase-9 (MMP-9), the reduction of protein adducts formation and the reduction of filaggrin loss, which are induced by dermal exposure to diesel engine exhausts (DEE). The pharmacological basis of the anti-inflammatory action of carnosol and carnosic acid from *Rosmarinus officinalis* was first investigated in vivo by Mengoni et al., who evaluated how these bio-active compounds could modify the expression of the inflammation-associated genes cyclooxygenase-1/2 (COX-1, COX-2), interleukin 1β (IL-1β), intercellular adhesion molecule 1 (ICAM-1), tumor necrosis factor-α (TNF-α) and fibronectin, thus revealing a downregulation of IL-1β and TNF-α, leucocyte migration reduction and, most interestingly, selective COX-2 inhibitory activity [47]. Regarding the potential use of *Rosmarinus officinalis* in atopic dermatitis, beyond the first study that more generally attributed an ameliorative role in skin lesions to nerve growth factor (NGF) inhibition, a more recent and detailed molecular characterization of carnosol’s intracellular action partially recalled the already known molecular targets and brought to light new ones involved in the implementation of its antiphlogistic action [48]. The most interesting aspect that emerged was the interaction of carnosol with the signal transducer and activator of transcription 3 (STAT3) pathway, which promotes skin inflammation via the upregulation of inducible nitric oxide synthase (iNOS) and COX-2. The carnosol–STAT3 interaction blocks lipopolysaccharide (LPS)-induced STAT3 phosphorylation. On the other hand, carnosol also showed a direct downregulation of the LPS-induced expression of iNOS, COX-2 and nitric oxide (NO). Such synergistic effects of this phenol explain its curative potential in AD [49]. Further confirmation of these molecular mechanisms comes from an in vivo study conducted by Yeo et al., highlighting the anti-inflammatory role of carnosol in UVB-induced skin damage in AD-affected mice. The topical application of carnosol reduced epidermal thickness, erythema, edema and erosion, and dramatically decreased serum levels of the pro-inflammatory cytokines TNF-α and IL-1β, together with a significant reduction in UVB-induced serum IgE [50].

A much more widely represented topic in the literature is the photoprotective role of *Rosmarinus officinalis* and its bioactive elements in UVR-induced skin aging. Martin et al. identified the photoprotective role of a water-soluble extract of *Rosmarinus officinalis* which downregulates both the basal levels of matrix metalloproteinase-1 (MMP-1) and the transcription of UVA/UVB-induced MMP-1 in dermal fibroblasts. Moreover, in a reconstructed skin model, the MMP-1 downregulatory action was also demonstrated further upstream, through the decrease in the UV-induced cytokines IL1- α and IL-6 [51].

In the wake of the above-mentioned work, Park et al. deeply evaluated the molecular mechanisms underlying the anti-photoaging activity of carnosic acid, identifying its ability to downregulate the UV-induced expression of MMP-1, MMP-3 and MMP-9 in human fibroblasts and keratinocytes. In addition, its UVB-induced MMP expression inhibitory activity would recognize the carnosic acid-mediated suppression of MEK/ERK/AP-1 pathways as an upstream causative agent [52]. The photoprotective action on the skin is not only a prerogative of the active bioelements of *Rosmarinus officinalis*, but also involves compounds from other plants, such as citrus flavonoids, in a synergic action that is even stronger than the therapeutic efficacy of the plants taken individually. To this end, Pérez-Sánchez et al., in their clinical trial, highlighted the advantageous effects of combinations of these extracts in human HaCaT keratinocytes, a spontaneously transformed aneuploid immortal keratinocyte cell line from adult human skin, and in human volunteers after oral supplementation. A consistent reduction in the UVB-induced formation of ROS and prevention of DNA damage was demonstrated in HaCaT cells, with consequent higher survival. In human volunteers taking 250 mg of combined citrus and rosemary extracts daily, a significant and progressive increase in minimal erythema dose (MED) was detected, thus suggesting that oral supplementation could improve UVB protection [53]. Further confirmation of the synergistic photoprotective, anti-inflammatory and anti-aging efficacy of these plants came a few years later from Nobile et al., who showed both in a human cell model and in a pilot study beneficial effects such as reduced UVR-induced erythema, diminished skin lipoperoxides (LPO), decreased wrinkle depth and improved elasticity following long-term oral supplementation of combined extracts of *Rosmarinus officinalis* and *Citrus paradisi*. These effects are probably attributable to the inhibition of UVR-induced ROS formation, as well as proinflammatory cytokines, coupled with the downregulatory action of intracellular matrix metalloproteinase-activating pathways [54].

The recent literature is teeming with works confirming the photoprotective role of various plant extracts, of which rosemary, with its bioactive elements, is an increasingly consistent member. Hyuck Auh et al. pioneered the investigation of the anti-photoaging potential of combined extracts of marigold and rosemary, finding in a mouse model that the oral supplementation of these extracts suppressed UV-induced dermal–epidermal thickening in a dose-dependent manner; this histological finding was supported by reductions in various photoaging-related biomarkers observed in the lab [55]. A new molecular photoprotection mechanism was recently highlighted by Calniquer et al., who studied the phytotherapeutic efficacy of a combination of tomato and rosemary extracts. They found an interesting functional synergy of polyphenols (mainly represented by carnosic acid and carnosol in the rosemary extract) and carotenoids (mainly represented by lycopene, phytoene and phytofluene in the tomato extract) in activating the antioxidant response element/Nrf2 (ARE/Nrf2) transcription system, the main cellular antioxidant defense mechanism, in parallel with the inhibition of the UVB-induced pro-inflammatory nuclear factor kappa B (NFκB) pathway in keratinocytes and dermal fibroblasts, resulting in a decreased release of IL-6 and TNF-α and consequently a lowered activation of MMPs [56]. Additionally, for photoprotective purposes, the most recent strand of cosmetic research is working on the development of topical gel formulations containing *Rosmarinus officinalis* extract. The antioxidant, antiaging and healing potential of rosemary hexane extract was evaluated both in vitro and in a UVB-irradiated mouse model. Ibrahim et al. demonstrated the photoprotective potential of rosemary extract, the permeability and bioavailability of which improved when topically conveyed into lipid nanocapsule-based gel [57]. In the wake of the aforementioned work, Takayama et al. evaluated the in vitro antioxidant potential of rosemary hydroethanolic extract together with an evaluation, in vivo, of its anti-UVB photoprotective role if topically conveyed by an emulgel formulation [58]. The main findings regarding the role of *Rosmarinus officinalis* and its antioxidant functions are summarized in Table 1. The main antioxidant properties of *Rosmarinus officinalis* against UV-induced and pollution-induced skin aging and against cutaneous inflammation are shown in Figure 1.

### 2.2. Rosmarinus Officinalis and Antimicrobial Action

In recent times, the scientific literature has provided ever-increasing knowledge on the antimicrobial activities of essential oils, which are finding use in both medical and cosmetic fields. Secondary plant metabolites have a pharmacological effect on the treatment of skin disorders, for which they are used within topical formulations. In this regard, rosemary extract has shown antimicrobial activity in several cases [59,60,61,62]. Starting from the assumption that EOs possess antimicrobial activity, De Macedo et al. reported the first attempt to produce an oil-in-water emulsion containing only natural excipients and rosemary extract, demonstrating that higher quantities of extracted phenolic compounds, flavonoids and tannins corresponded to greater antioxidant and antimicrobial activity, especially against *Staphylococcus aureus, Streptococcus oralis* and *Pseudomonas aeruginosa*. De Macedo et al. concluded by asserting that this topical formulation based on *Rosmarinus officinalis* could be a natural therapeutic novelty against microorganisms, which are becoming increasingly resistant to conventional drugs [63]. Another field of research is the treatment of Candida species, which are developing increasing resistance to traditional drugs [64,65], including azoles and polyenes [66,67], thus representing a great challenge for the medical field in the treatment of such common skin infections. Furthermore, conventional antifungal drugs often exhibit toxic side effects for human cells, including hepatotoxic effects. Hence, there is growing scientific interest in EOs, natural compounds that are proving to be promising for their various antibacterial, antifungal and antiviral properties [68,69]. Their action against a wide variety of microorganisms is believed to be due to their ability to alter the membrane and the microorganisms’ cell wall, resulting in the extracellular loss of cytoplasmic material [70]. The pharmacological properties of EOs, mainly related to their complex chemical makeup and high levels of phenols, make these compounds a promising tool for the treatment and prevention of candidiasis [71]. However, the low solubility in water, high volatility and high instability of EOs represent the main limitations of their use in the pharmaceutical and cosmetic fields. To overcome this problem, the encapsulation of EOs has proven to be a useful solution. Starting from the knowledge that *Rosmarinus officinalis* can be successfully used as a matrix component and active ingredient of nanostructured lipid carriers (NLCs), it was subsequently demonstrated that the nanoparticles ensured a prolonged in vitro release of clotrimazole, thus increasing the antifungal activity. This confirmed that NLCs containing Mediterranean EOs represent a promising strategy to improve efficacy against topical candidiasis [71]. Specifically, an in vitro test highlighted that leaves from *Rosmarinus officinalis* and *Tetradenia riparia* contain antifungal bioactive compounds. Hydro-alcoholic extracts from these leaves seem to be effective against dermatophytes, including *Trichophyton rubrum, Trichophyton mentagrophytes* and *Microsporum gypseum*, with fungal growth inhibition and morphological alterations in the hyphae [72]. Moreover, it has been proven that the aryl hydrocarbon receptor (AhR), when activated by microbial metabolites, is implicated in a number of skin diseases. Starting from this assumption, Kallimanis et al. attempted to identify natural compounds potentially capable of inhibiting AhR activation by microbial ligands. In this regard, five different dry extracts of *Rosmarinus officinalis* were analyzed to evaluate its ability to inhibit AhR, confirming its dose-dependent antimicrobial activity against *Malassezia furfur* [73]. Another study conducted by Weckesser et al. evaluated the antimicrobial capacity of six extracts, including *Rosmarinus officinalis*, and plant extracts, including carnosol and carnosic acid, showing that *Rosmarinus officinalis* extract inhibited the growth of both aerobic and anaerobic bacteria and yeasts. Furthermore, the Rosmarinus extract was able to inhibit both Candida strains, further showing its antifungal action [74]. Sporotrichosis is a subcutaneous fungal infection caused by fungi of the genus *Sporothrix*, increasingly affecting humans and cats [75]. Despite the limited experimental data on the effects of rosemary essential oil on the treatment of sporotrichosis, studies in the current literature confirm its antifungal activity against *Sporothrix schenckii* [76] and itraconazole-resistant *Sporothrix brasiliensis* [76,77]. The first attempt to assess the effectiveness of rosemary essential oil against cutaneous sporotrichosis in vivo was conducted by Waller et al. Itraconazole-resistant *Sporothrix brasiliensis* was inoculated into 30 Wistar mice, which were randomly treated with itraconazole, rosemary oil or saline as the control population. In mice treated with rosemary oil, the remission of skin lesions was noted, with mild to absent yeast cells. Furthermore, rosemary oil has also shown a protective effect on systemic organs, such as the liver and spleen, delaying the spread of infection [78]. Finally, rosemary has been evaluated for its antibacterial functions. It is known that bacterial pathogens have numerous virulent mechanisms allowing them to enter, replicate and persist at host sites, but with only a few common mechanisms. Among the possible alternatives to overcoming the issue of the constant increase in antibiotic resistance, inhibiting the virulence factors, which are responsible for the damage caused to the host tissue, is increasingly gaining ground as a new line of research [79]. By specifically inhibiting bacterial virulence mechanisms, the pathogenicity of bacteria could be controlled, thus avoiding the increasing ability of bacteria to develop antimicrobial resistance. Furthermore, selectively inhibiting virulence mechanisms reduces the risk of altering the composition of commensal microorganisms, which also play a beneficial role within and on the host. Quorum sensing, as a virulence mechanism, is a cell-density-dependent transcriptional regulatory system, used by bacteria to communicate and to adapt to the environment [80]. *Staphylococcus aureus*, a common cause of skin and soft tissue infection (SSTI), has generated increasing concern due to drug resistance. *Staphylococcus aureus* is an opportunistic Gram-positive bacterium, whose virulence mechanisms involve the activation of the quorum-sensing accessor gene regulator (agr) operon. The diterpene carnosic acid and carnosol, found in *Rosmarinus officinalis L*. leaves, have been demonstrated to have a specific inhibitory effect on *Staphylococcus aureus* agr expression, thus suppressing the cell–cell communication system and, consequently, its pathogenicity [80]. Finally, the activity of basil and rosemary essential oils has also been demonstrated against multi-resistant clinical strains of *Escherichia coli*. The results show that both essential oils tested were active against all clinical strains of *Escherichia coli*, including broad-spectrum β-lactamase-positive bacteria [81]. All of this evidence supports the idea that *Rosmarinus officinalis* can be used both as an important therapeutic tool and as an adjuvant within cosmetological formulations for its broad-spectrum antioxidant and antimicrobial capacities. Table 2 summarizes the main findings about the antimicrobial activity of *Rosmarinus officinalis*.

### 2.3. Rosmarinus Officinalis and Wound Healing

In the past decade, a new and interesting strand of phytotherapeutic research has become increasingly popular, which aims to evaluate the curative potential of *Rosmarinus officinalis* in wound healing, as well as in promoting the survival of skin flaps. Concerning the wound-healing process, beyond some in vivo evidence of the regenerative potential of *Rosmarinus officinalis* in the treatment of acute wounds, especially burn wounds, much of the scientific evidence instead concerns the therapeutic potential of rosemary in chronic wounds, primarily diabetic wounds [82,83].

First, Abu-Al-Basal, in an in vivo study conducted on BALB/c mice, demonstrated the efficacy of both aqueous extract and essential oil of *Rosmarinus officinalis* in healing diabetic wounds by pointing out the greater healing efficacy of the essential oil over the aqueous extract [84]. In the wake of this phytotherapeutic interest, Sivamani et al. evaluated the wound-healing role of EO components from various plants, identifying as a potential molecular therapeutic mechanism their ability to inhibit elastases produced by skin, neutrophils and germs, including Pseudomonas aeruginosa [85]. Additionally, Pérez-Recalde et al. confirmed the therapeutic potential of EOs of various plants, including rosemary, especially in chronic wounds. In rodent wounds, improved collagen deposition associated with increased fibroblastic proliferation and a faster wound closure rate has been observed, even highlighting the promising role in wound healing of the incorporation of EOs into resorbable polymeric scaffolds [86]. Similarly, Labib et al. highlighted in vivo the wound-healing potential of a combination of rosemary and tea tree essential oils incorporated into chitosan-based preparations. In the excision wound model in rats, their topical application resulted histologically in complete re-epithelialization associated with follicular activation, together with a significant increase in the rate of wound contraction. In addition, a marked reduction in oxidative stress in the wound area was highlighted, probably attributable to the antioxidant capacity of oxygenated monoterpenes, well-represented in both essential oils examined [87]. The most recent in vivo evidence on the wound-healing potential of rosemary highlights the anti-fungal role that bioactive compounds in its EO such as α-pinene might play, thereby speeding up the healing process [88].

Regarding the potential use of *Rosmarinus officinalis* in improving skin flap survival in relatively recent times, Ince et al. topically tested this ability in vivo with encouraging results [89]. These last were soon confirmed by the same author in another in vivo study that highlighted the vasodilatory effect of orally administered *Rosmarinus officinalis* oil. The resulting increased blood flow to the flap averted the dreaded necrotic complication, suggesting a systemic use of rosemary, especially in patients with circulatory disorders such as chronic obliterative artery disease [90]. In line with the future goals of the aforementioned work, in their latest in vivo study, Ince et al. identified two bioactive compounds from the essential oil of *Rosmarinus officinalis*, alpha-pinene and cineole, as the main systemic contributors to the increased flap survival [91]. Table 3 summarizes the main findings regarding the role of *Rosmarinus officinalis* in wound healing and skin flap survival.

### 2.4. Rosmarinus Officinalis and Cutaneous Diseases

Rosemary has been shown to have not only antioxidant and antimicrobial properties, but also a beneficial role in the treatment of various skin diseases.

In the treatment of alopecia aerate, an autoimmune disease affecting the follicles with subsequent hair loss, the essential oil of *Rosmarinus officinalis* managed to improve microcirculation surrounding the hair follicle [92]. Moreover, a clinical study compared the efficacy of rosemary essential oil to minoxidil 2% solution for the treatment of androgenetic alopecia. Patients used either minoxidil 2% solution or rosemary essential oil, with a dramatic increase in hair count reported for both treatments and without significant differences between the two study groups, which confirmed the therapeutic effectiveness of *Rosmarinus officinalis*. Moreover, scalp irritation was more frequent in the minoxidil 2% solution group, confirming the relatively few side effects of natural compound therapies [93]. Among the possible therapeutical approaches, increasing attention has been paid to platelet-rich plasma (PRP) to increase hair density and regrowth. The effect of PRP in combination with herbal extracts has been evaluated to identify the factors stimulating hair growth [94]. Combined herbal extracts and PRP promoted the proliferation of human dermal papilla cells via the regulation of extracellular signal-regulated kinase (ERK) and protein kinase B (Akt) proteins, shedding light on the possible future development of herbal extracts and PRP combination therapies in order to enhance hair growth. The role of *Rosmarinus officinalis* has also been evaluated in systemic sclerosis-related Raynaud’s. Additionally, in this connective tissue disorder, the high levels of ROS contribute to the development of fibrotic processes and closely correlate with the severity of skin fibrosis [95,96]. In an open-label pilot study, Vagedes et al. enrolled twelve patients, each of whom received an application of olive oil on both hands as a control and three hours later an application of 10% essential oil of *Rosmarinus officinalis L*., highlighting that warmth perception in patients with Raynaud’s phenomenon was ameliorated by topical rosemary EO application [97]. The potential role of rosemary has also been evaluated from an aesthetic perspective, specifically in the treatment of cellulite, a condition characterized by localized adiposity and inflammation, with subsequent alteration of the microcirculation, mostly affecting women. On this topic, in 3T3-L1 cells (a mouse cell line with an adipocyte-like phenotype), a composition of extracts, including those from *Rosmarinus officinalis*, was shown to reduce lipid accumulation, platelet aggregation and inflammation, thus ameliorating microcirculation through a dose-dependent inhibition of free radical formation. This evidence suggests the potential topical use of rosemary, combined with other extracts, and also in the aesthetic field, taking advantage of its anti-inflammatory and antioxidant properties [98]. As already discussed, overexposure to UVB rays causes oxidative stress and DNA damage, resulting in an increased likelihood of developing different types of cutaneous cancer, including non-melanoma skin cancer and malignant melanoma [99,100]. ROS plays a pivotal role in oncogenesis and mutagenesis, especially in tumor promotion. ROS induces lipid peroxidation and DNA strand breaks by modulating different biochemical pathways and gene expression [101]. In recent years, increased scientific attention has been paid to identifying and characterizing natural compounds with chemopreventive properties against the formation of UVB-induced skin cancer [102].

On this topic, the potential role of carnosol in the chemoprevention of UVB light-induced non-melanoma skin cancer has been evaluated in an HaCat cell study, highlighting that carnosol leads to a partial reduction in UVB-induced ROS and subsequently in a reduction in DNA damage. This ability consists of the absorption of UVB radiation, which in turn could decrease the UVB-induced formation of cyclobutane pyrimidine dimers (CDP) in keratinocytes, further inhibiting the UVB-induced activation of NFκB and UVB-induced mutation [103].

In a mouse model, the modulatory effects of *Rosmarinus officinalis* were studied by *evaluating* the *skin tumor mean latency period*, incidence, burden, yield, weight and diameter. Mouse skin carcinogenesis, evaluated by the formation of papillomas, was induced by topical application on the dorsal skin of 7,12-dimethlybenz(a)anthracene (DMBA) and promoted by croton oil. The results of the study conducted by Sancheti et al. suggest that *Rosmarinus officinalis* leaf extract could postpone the onset of papillomas and their latency period. Furthermore, confirming its antioxidant action, it was observed that serum levels of lipid peroxidation, an index of cellular oxidation, were significantly reduced in mice treated with *Rosmarinus officinalis* [104]. 

A growing body of evidence indicates that specific compounds of rosemary, including carnosol, carnosic acid and rosmarinic acid, exert antiproliferative activity in several cancer cell lines [105,106,107,108]. In colorectal cancer cells, rosmarinic acid causes apoptosis [109], downregulating the mitogen-activated protein kinase (MAPK)/ERK pathway, while in hepatocellular carcinoma cells, rosemary essential oil reduced bcl-2 gene expression and upregulated bax gene expression [110]. In vivo, the anticancer properties of rosemary were proven in mice with acute myeloid leukemia, in which the increase in the administration of crude extracts of rosemary or carnosol, in combination with 1α-25 dihydroxy vitamin D3, led to an intense cytoprotective effect [111]. Thus, in vitro and in vivo data also indicated that crude extracts or purified components of rosemary exerted chemoprotective effects, inhibiting the early stages of tumor development [112,113], probably through the inhibition of enzymes of stage I carcinogenesis. Among the tumors with a rapidly increasing incidence rate, melanoma is a malignant tumor induced by the transformation of melanocytes [114]. When metastatic, the prognosis of melanoma becomes very bad, especially due to the poor response to the currently approved therapies. Hence, the growing interest in EOs is justified. On this topic, Huang et al. demonstrated in vitro that carnosol inhibited the migration of metastatic B16/F10 mouse melanoma cells through the suppression of MMP-9 expression. Furthermore, carnosol was shown to inhibit ERK1/2, AKT, p38 and c-Jun N-terminal kinases (JNK), and led to the activation of the transcription factors NFκB and c-Jun. From this assumption, the authors concluded that the invasive capacity of B16/F10 mouse melanoma cells could be limited by carnosol, through the downregulation of the above-mentioned pathways [115]. Finally, Cattaneo et al. highlighted that the proliferation of human melanoma cell line A375 was reduced by the hydroalcoholic extract of *Rosmarinus officinalis*, in a dose- and time-proportional way through cytotoxic and cytostatic effects on the cell cycle. Through the compositional characterization, the individual pure components of the extract were tested. The observations led researchers to hypothesize that the antiproliferative activity was a property of the entire extract, most likely deriving from multifactorial effects involving the majority of its elements [116]. All of these data agree in stating the potential role of rosemary in the therapy of various skin pathologies, first of all among skin cancer. From the analyzed studies, it emerges that the anti-cancer action derives in the first instance from its antioxidant action, which in its turn inhibits the genesis and progression of the tumor. Table 4 summarizes the main findings regarding the role of R. officinalis in cutaneous diseases.

### 2.5. Rosmarinus Officinalis and Cutaneous Lymphoma

Other skin disorders, including lymphomas, may benefit from the antioxidant properties of rosemary. A rare and frequently severe T-cell lymphoma, which can develop in the blood, lymph nodes or skin, is known as adult T-cell leukemia/lymphoma (ATLL). Human T-cell lymphotropic virus type 1 (HTLV-1) infection has been related to ATLL onset; however, less than 5% of HTLV-1 infected-patients develop ATLL. The Caribbean, some regions of South and Central America, and some portions of Africa are the areas where the HTLV-1 virus is most prevalent. To date, we are unable to predict which infected patients will develop ATLL. Through their crucial functions in accelerating cell proliferation and preventing cell death, the viral genes tax and HTLV-1 bZIP factor (HBZ) supports the growth of infected cells. An ATL clone emerged as a result of the persistence of infected clones in vivo and the accumulation of genetic mutations and abnormal epigenetic alterations in host genes [117]. According to a study, the viral oncoproteins Tax and HBZ generate oxidative stress, mitochondrial damage and cytotoxicity, which are countered by the TP53-induced glycolysis and apoptosis regulator (TIGAR), which in turn is induced by the HTLV-1 latency-maintenance factor p30II. In colony transformation and foci formation assays, the p30II protein works in concert with Tax and HBZ to increase their oncogenic potential [118]. Additionally, in an in vivo xenograft model of HTLV-1-induced T-cell lymphoma, the authors demonstrated that TIGAR is substantially expressed in HTLV-1-induced tumors linked to oncogene deregulation and enhanced angiogenesis. These results show that the key oncoproteins Tax and HBZ likely work together as cofactors during retroviral carcinogenesis [119]. Therefore, reducing oxidative stress could alter the proliferative dynamics in ATLL patients. An experimental study revealed that carnosol caused ATL cell apoptosis through the inhibition of cell proliferation. The authors then used mass spectrometry and proteome analysis with fluorescent two-dimensional differential gel electrophoresis to look into the apoptosis-inducing mechanism of carnosol. According to the proteome study, carnosol-treated cells expressed more reductases, glycolytic pathway enzymes and enzymes in the pentose phosphate pathway than untreated cells did. These findings suggest that carnosol had an impact on the cell redox state. Additionally, the quantitative examination of glutathione, which is crucial for maintaining the intracellular redox state, revealed that carnosol was the reason for the decreased glutathione levels in cells. Furthermore, N-acetyl-L-cysteine, which is the precursor of glutathione, reduced carnosol efficiency. From these findings, it was suggested that the apoptosis-inducing activity of carnosol in ATL cells was provoked by the depletion of glutathione [120]. Although the results in the literature on the relationship between *Rosmarinus officinalis* and cutaneous lymphomas are rather limited, in vitro studies would seem to confirm the antineoplastic activity of this substance. Visanji et al. studied the antiproliferative effects of carnosol and carnosic acid on Caco-2 cells, demonstrating that after incubation with these components, the cells increased their doubling time, i.e., the time required to double their population. This was estimated to be due to G2/M phase cell cycle arrest. Furthermore, carnosic acid and carnosol were observed to arrest the cell cycle at different times. While carnosic acid arrested cells before prometaphase by reducing cyclin A levels, carnosol exerted its major impact on the cell cycle after prometaphase [121]. All of these data could provide the basis not only for an investigation of the potential chemopreventive role of Rosmarinus through cell cycle arrest, but also for an evaluation of the existence of the possible synergistic action of rosemary with traditional chemotherapeutic drugs, to assess the possibility that it can reduce the evolution of viral infection to neoplastic disease.

## 3. Materials and Methods

This research was carried out on the PubMed database, using the keywords “Rosmarinus officinalis” and “skin”. The preliminary research excluded previous reviews and systematic reviews, along with articles not in the English language. The results were screened and selected in the following order: title, abstract and content. Double results were screened and removed from the final article count. Table 1, Table 2, Table 3 and Table 4 report the articles that were included and reviewed, divided into sections by topic.

## 4. Conclusions and Future Perspectives

Ever-increasing scientific attention is being paid to phytotherapeutics, plants with potential therapeutic activities. Among these, *Rosmarinus officinalis L*., a medicinal plant native to the Mediterranean region, already well-known and thoroughly investigated for its anti-cancer potential, is increasingly showing promising results for its antioxidant and anti-inflammatory activity due to the interaction between the bioactive elements of the plant and the molecular pathways governing inflammatory processes, as well as the pro-oxidative/antioxidant balance. Among the organs that would benefit from such healing effects, according to the evidence, the skin stands out. In light of the promising and well-documented in vitro effects and in vivo results in animals, the future goals of phytotherapeutic research should be geared towards an expansion of clinical trials, especially aimed at investigating in more depth the single bioactive elements of the plant and characterizing new ones, as well as assessing their therapeutic efficacy when combined with other plant extracts. Another pursuable goal, with a view to systemic use, concerns the need for more studies to establish therapeutic dosages of this plant or its bioactive elements. For the latter, the development of innovative technologies directed at their targeted extraction should be encouraged in order to meet the growing future demand for phytotherapy. A further critical issue to be investigated considering the increasingly confirmed recognition of phytodermatitis as a clinical entity [122] is the need for a more in-depth assessment of the risk–benefit balance linked to the topical use of the plant. From what has been said so far, although the use of rosemary in the treatment of skin diseases represents a fascinating line of research, future perspectives still require large and controlled clinical trials in order to definitively elucidate the real impact of this plant and its components in clinical practice.

## Figures and Tables

**Figure 1 antioxidants-12-00680-f001:**
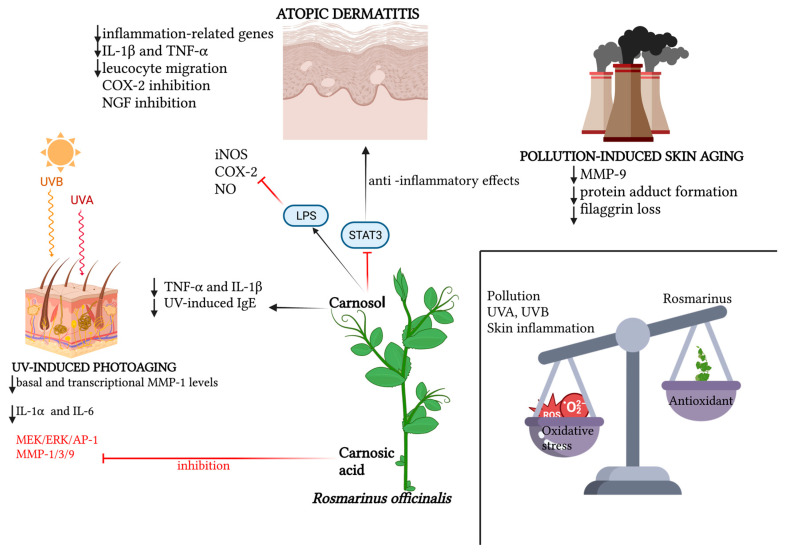
Schematic representation of the relevant molecular patterns involved in the promising antioxidant effects of *Rosmarinus officinalis* and its main bioactive compounds on three dermatological conditions: atopic dermatitis, UV-induced photoaging and pollution-induced skin aging. In atopic dermatitis, beyond the synergic action of carnosol and carnosic acid in downregulating inflammation-related genes, and therefore pro-inflammatory cytokines, leucocyte migration and NGF inhibition promotion, the specific action of carnosol on the STAT3 pathway is emphasized, for which the LPS-induced phosphorylation lock results in anti-inflammatory effects, together with carnosol’s direct inhibition of iNOS, NO and COX-2 activation. In UV-induced photoaging, the action of *Rosmarinus officinalis* in downregulating basal and transcriptional levels of MMP-1, as well as the inflammatory cytokines IL-6 and IL-1α, is highlighted. Also portrayed is the specific action of carnosol in downregulating TNF-α, IL-1β and serum levels of UV-induced IgE, and the inhibitory action of rosmarinic acid both on the MEK/ERK/AP-1 pathway and on MMP-1, MMP-3 and MMP-9. In pollution-induced skin aging, the downregulatory action of the two main phenolic diterpenes of *Rosmarinus officinalis* is mainly demonstrated on MMP-9, protein adduct formation and the loss of filaggrin. Created with BioRender.com.

**Table 1 antioxidants-12-00680-t001:** The role of *Rosmarinus officinalis* against oxidative stress.

Authors and Year	Topic	Model	Extraction Procedure	Study Characteristics
Takayama et al. [58], 2022	Antioxidants and UVB protection	In vivo/in vitro	Exhaustive maceration	An in vitro and in vivo study on the properties of R. officinalis demonstrated its protective role for the skin against tissue damage caused by UVB radiation.
Nobile et al. [54], 2016	Antioxidants and UVR protection	In vivo	Drying	The antioxidant, photoprotective and antiaging efficacy of the combination of *Rosmarinus officinalis* and *Citrus paradisi* extracts was demonstrated.
Ibrahim et al. [57], 2022	Antioxidants and anti-aging	In vivo/in vitro	Not specified	The photoprotective potential of rosemary extract, whose permeability and bioavailability improved when topically conveyed into lipid nanocapsule-based gel, was assessed.
Nobile et al. [45], 2021	Antioxidants and pollution	In vivo	Not specified	A double-blind randomized study demonstrated that oxidative stress-induced skin damage in both Asian and Caucasian women living in a polluted urban is reduced by oral supplementation with the following herbal extracts: *Olea europaea* leaf, *Lippia citriodora, Rosmarinus officinalis,* and *Sophora japonica.*
Mengoni et al. [47], 2011	Antioxidants, inflammation and AD	In vivo/in vitro	Drying	In a mouse model, the expression of IL-1β and TNF-α, markers of inflammation-associated genes in skin, were reduced by carnosic acid and carnosol.
Calniquer et al. [56], 2021	Antioxidants and UVB protection	In vitro	Not specified	An in vitro study demonstrated that the combination of carotenoids and polyphenols produces protective effects against UV-induced damage to skin cells, inhibiting UVB-induced NFκB activity and IL-6 release.
Sanchez et al. [53], 2014	Antioxidants and UVB protection	In vivo/in vitro	Drying and water dissolution	In HaCaT keratinocytes and in human volunteers, the oral intake of rosemary and citrus bioflavonoid extracts reduced UVB-induced ROS, thus preventing cellular DNA damage.
Kim et al. [42], 2003	Antioxidants	In vitro	Ethanol/water (50:50, *v*/*v*)	The protein glycation inhibitory activity of aqueous ethanolic extracts of various plants, including *Rosmarinus officinalis*, closely correlated with the antioxidant activity of the extracts.
Salem et al. [44], 2020	Antioxidants	In vitro	Drying, maceration, water distillation, boiling, filtration, lyophilization	An in vitro study evaluated the radical-scavenging and anti-aging activity of aqueous and ethanoic extracts of phenolic-rich selected herbs, including *Rosmarinus officinalis*, which showed the highest antioxidant activity and the most pronounced anti-elastase, anti-tyrosinase and anti-collagenase activity.
Ezzat et al. [43], 2016	Antioxidants	In vivo/in vitro	Drying, pulverization, defatting, percolation with 70% ethanol, evaporation	The anti-wrinkle activity of DER, which was optimized by encapsulation in transferosomes, was assessed in an in vitro study.
Yeo et al. [50], 2019	Antioxidants and atopic dermatitis	In vivo	Not specified	The anti-inflammation effect of the topical application of carnosol on UVB-induced skin inflammation in HR1 mice inhibited erythema, epidermal thickness and inflammatory responses.
Lee et al. [49], 2019	Antioxidants and atopic dermatitis	In vivo/in vitro	Not specified	Carnosol inhibited LPS-induced nitric oxide generation and the expression of inflammatory marker proteins, including iNOS and COX-2 in RAW 264.7 cells. STAT3 phosphorylation and DNA-binding activity in RAW 264.7 cells were reduced.
Takano et al. [48], 2011	Antioxidant and atopic dermatitis	In vivo/in vitro	Ethanol	In an atopic dermatitis mouse model, the application of four herbal extracts, including *Rosmarinus officinalis*, reduced atopic lesions, thus inhibiting the effect of NGF on neuritic outgrowths in lesional skin.
Martin et al. [51], 2008	Anti UV	In vitro	Solubilization	IL1-α and IL-6, which play a role in the up-regulation of UV-induced MMP-1, could be suppressed by the *Rosmarinus officinalis* water-soluble extract.
Park et al. [52], 2013	Anti UV	In vitro	Not specified	The antiaging activity of carnosic acid downregulated the UV-induced expression of MMP-1, MMP-3 and MMP-9 in human fibroblasts and keratinocytes.
Hoskin et al. [46], 2021	Antioxidants and pollution	Ex vivo	Hydroalcoholization	The topical application of a gel based on hydroalcoholic rosemary extract complexed with algae proteins against pollution-induced oxidative skin damage was demonstrated.
Hyuck Auh et al. [55], 2021	Anti UV	In vivo	Ethanol at 72 °C for 3 h, evaporation	A mixture of marigold and rosemary extracts demonstrated anti-aging activity in a UV-induced mouse model of photoaging, with reduced expression of matrix metalloproteinase, interleukins, TNF-α, procollagen type I, superoxide dismutase, glutathione peroxidase and catalase

**Table 2 antioxidants-12-00680-t002:** The antimicrobial activity of R. officinalis.

Authors and Year	Topic	Model	Extraction Procedure	Study Characteristics
Kallimanis et al. [73], 2022	Anti-microbial activity	In vitro	After drying, the leaf was treated with each solvent in a 1:10 ratio, and then separated from the liquid part by filtration.	Five different dry ROEs were assayed for their activities as antagonists of AhR ligand, which in turn inhibited *Malassezia furfur* yeasts.
De Macedo et al. [63], 2022	Anti-microbial activity	In vitro	Maceration, infusion, Soxhlet and ultrasound	A topical formulation with R. officinalis extract demonstrated antimicrobial activity against *S. aureus, S. oralis,* and *P. aeruginosa*
Endo et al. [72], 2015	Anti-microbial activity	In vitro	Leaf were dried in a circulating-air oven at 40 °C. Subsequently, they were soaked in 90/10% (*v*/*v*) ethanol–water for 48 h at 25 °C, protected from light.	Hydroalcoholic extracts from R. officinalis and T. riparia in vitro was demonstrated to have antifungal activity against strains of *Trichophyton rubrum, Trichophyton mentagrophytes* and *Microsporum gypseum*
Nakagawa et al. [80], 2020	Anti-microbial	In vitro	Not specified.	Diterpene carnosic acid and carnosol, present in *Rosmarinus officinalis L*. leaves, had specific effect on S. aureus agr expression.
Waller et al. [76], 2021	Anti-microbial activity	In vitro	Distillation by steam dragging in Clevenger equipment for 4 h	The study demonstrated rosemary oil as a promising antifungal to treat sporotrichosis, thus postponing systemic fungal spreading.
Weckesser et al. [74], 2007	Anti-microbial activity	In vitro	The solvent used was Carbon dioxide/isopropyl alcohol.	Rosmarinus extract inhibited the growth of *Candida* strains
Sienkiewicz et al. [81], 2013	Anti-microbial activity	In vitro	Not specified.	Basil and *Rosmarinus officinalis* essential oils played a role against resistant *Escherichia coli* clinical strains, and also against extended-spectrum β-lactamase positive bacteria.
Carbone et al. [71], 2013	Anti-microbial activity	In vitro	Not specified.	Nanostructured lipid carrier systems containing EOs, including *Rosmarinus officinalis*, could improve Clotrimazole effectiveness against candidiasis.

**Table 3 antioxidants-12-00680-t003:** The role of R. officinalis in wound healing and skin flap survival.

Author	Topic	Model	Extraction Procedure	Study Characteristics
Labib et al. [87], 2019	Wound Healing	In vivo	Not specified	The wound-healing potential of a combination of rosemary and tea tree essential oils incorporated into chitosan-based preparations was highlighted.
Abu-Al-Basal et al. [84], 2010	Wound healing	In vivo	Steam distillation	An in vivo study conducted on BALB/c mice demonstrated the efficacy of both aqueous extract and essential oil of *Rosmarinus officinalis* in healing diabetic wounds.
Mekkaoui et al. [83], 2021	Wound healing	In vivo	Not specified	A honey mixture with selected essential oils on chemical and thermal wound models in rabbits has healing effects.
Sivamani et al. [85], 2012	Wound healing	In silico	Not specified	Rosmarinus, among other essential oils, inhibited the deleterious activities of elastase, thus ameliorating wound healing.
Sakhawy et al. [88], 2023	Wound healing	In vivo	Not specified	Topical application of a mixture of essential oils, including *Rosmarinus officinalis*, had potential in healing wounds infected with *Candida albicans*.
Farhan et al. [82], 2021	Wound healing	In vivo	Methanol extraction	In vitro, the antifungal activities of *Rosmarinus officinalis* in wounds infected with *Candida albicans* was demonstrated.
Ince et a [90], 2016	Increasing skin flap survival	In vivo	Not specified	The vasodilatory effects of *Rosmarinus officinalis* contributed to increasing skin flap survival.
Ince et al. [89], 2015	Increasing skin flap survival	In vivo	Not specified.	*Rosmarinus officinalis* increased skin flap survival in a mouse model.
Ince et al. [91], 2018	Increasing skin flap survival	In vivo	Not specified	Alpha-pinene and cineole were the components of *Rosmarinus officinalis* responsible for increased flap survival.

**Table 4 antioxidants-12-00680-t004:** The main findings regarding the role of *Rosmarinus officinalis* in cutaneous diseases.

Author and Year	Topic	Model	Extraction Procedure	Study Characteristics
Panahi et al. [93], 2019	Alopecia			*Rosmarinus officinalis* improved microcirculation surrounding the follicle, with comparable results to topical Minoxidil 2% in hair regrowth in patients affected by androgenetic alopecia.
Rastegar et al. [94], 2013	Alopecia	In vitro	The herbs were dried, crushed, and passed through 80-mesh stainless-steel sieves and water was used as a base.	Herbal extract with *Rosmarinus officinalis* and PRP had a positive effect on hair regrowth, promoting the proliferation of human dermal papilla.
Vagedes et al. [97], 2022	Raynaud’s phenomenon	In vivo	Not specified.	In an open-label pilot study, warmth perception in patients with systemic sclerosis-related Raynaud’s phenomenon was increased by the application of topical rosemary essential oil.
Yimam et al. [98], 2017	Cellulite	In vitro	Dried rosemary leaf was extracted with an approximately 10-fold volume of 95% ethyl alcohol at 40 °C.	A composition of extracts, including those from *Rosmarinus officinalis*, reduced lipid accumulation, platelet aggregation and inflammation, thus ameliorating microcirculation through antioxidant activity
Tong et al. [103], 2018	Non-Melanoma skin cancer	In vitro	Not specified.	Carnosol inhibits the UVB-induced activation of NF-κB, thus reducing keratinocyte carcinogenesis in vitro
Sancheti et al. [104], 2006	Skin cancer	In vivo	Extraction in a Soxhlet apparatus with double-distilled water by refluxing for 36 h at 50–60 °C.	A mouse model demonstrated the protective role of *Rosmarinus officinalis* against skin tumorigenesis
Huang et al. [115], 2005	Melanoma	In vivo	Extraction with hexane, solvent evaporation, dissolving the dried material with methanol, and then filtrating and evaporating the solvent again.	Carnosol inhibited the migration of metastatic B16/F10 mouse melanoma cells in vitro by suppressing the expression of MMP-9
Cattaneo et al. [116], 2015	Melanoma	In vitro	Grinding into fine powder and suspension at 330 g/L in a solution of 65% (*w*/*w*) ethanol/water for 21 days. The extract was then filtered and stored at −20 °C until use.	In vitro, extract of *Rosmarinus officinalis L*. inhibited human melanoma A375 cell line proliferation in a dose- and time-proportional way

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
