# Peer review of "Rosmarinus officinalis and Skin: Antioxidant Activity and Possible Therapeutical Role in Cutaneous Diseases"

_antioxidants, 2023, doi:10.3390/antiox12030680_

Round 1
Reviewer 1 Report
This review on the effects of Rosmarinus officinalis extracts on skin diseases is interesting. The plant Rosmarinus officinalis is certainly promising to treat, prevent or complete another treatment of various skin diseases. However, as is the case in general in phytotherapy, the effects will depend on the extraction precedure and the vehicle used.
This could be indicated in the text; in Tables 1-4, it would be interesting to add columns with the model used and the extraction procedure, since the composition and the bioavailability of the products, as well as the transposition of the effects to in vivo human situation will depend on all these parameters.
It is important to have these informations when examining data from topical application of plant extracts, because the diversity of published results reflect the diversity of the procedures, the vehicles and the protocols used in the study.
Reviewer 2 Report
The manuscript „Rosmarinus officinalis and skin: antioxidant activity and possible therapeutical role in cutaneous diseases“ was submitted for revision and consideration of publication in the journal antioxidants. Federica Li Pomi and Vincenzo Papa share equally the authorship. The review manuscript contains an introduction, a main part, materials and methods and finally conclusions and future perspective. The main part in turn is divided into five chapters containing different fileds of interaction between rosmarinum officinalis or its active components and skin applications.
Based on the antioxidative properties of rosmarinum officinalis or its components an increasing attention emerged for its use in UV-damaged skin, anti-inflammatory or photoprotective activity. The authors provide a good overview on relevant publications and describe the findings in great detail. Further, rosmarinum officinalis or its components are described in regard to its antimicrobial activity, wound healing promoting activity, other cutaneus diseases and with a possible application for lymphoma. All aspects are supported and referenced by recent respective literature. While the first 3 chapters contain very specific molecular target molecules and pathways, the latter still requires further investigations and seems more like a conclusion by analogy. Fig. 1 summarizes graphically the described effects which helps to bring the facts together.
The approach and the topic are very interesting. The manuscript provides a good overview of the current state of the art on rosmarinum officinalis for applications in medicine and future opportunities. Occassionally, the sentences are very long and interleaved, making them difficult to follow.
Overall, I recommend using a professional English editing service. The entire manuscript must be checked for English grammar, spelling, and language.
In the following some minor points are addressed:
On p4 a study was cited that involved human volunteers as well as cultured keratinocytes. However the sentence (line 168-171) is confusing as it seems to mix up both. Please check.
Please use the same terms in the figure and legend text. i.e. Fig 1 UV-induced photoageing and UV-induced skin aging and pollution-induced skin damage and pollution-induced skin aging. This is just for a better recognition.
Line 413 „..skin tumour mean latency period…“… please add the specific type of skin cancer. Also in tabe 4 the type of skin cancer should be specified.
In chapter 4‚ ‚cutaneous disease‘ the disease Cellulitis was not discussed in the text but a reference is given in table 4. Please complete this part.
Round 2
Reviewer 2 Report
The authors considered all comments and amended the text accordingly. The manuscript was thouroughly revised.